# High resolution data modifies intensive care unit dialysis outcome predictions as compared with low resolution administrative data set

Jennifer Ziegler[1], Barret N. M. Rush[1], Eric R. Gottlieb[2,3,4], Leo Anthony Celi[3,4,5,6‡], Miguel Ángel Armengol de la Hoz[4,7,8‡] *

**1** Department of Internal Medicine, Max Rady College of Medicine, Rady Faculty of Health Sciences, University of Manitoba, Winnipeg, Manitoba, Canada, **2** Department of Medicine, Mount Auburn Hospital, Cambridge, Massachusetts, United States of America, **3** Harvard Medical School, Boston, Massachusetts, United States of America, **4** Institute for Medical Engineering and Science, Massachusetts Institute of Technology, Cambridge, Massachusetts, United States of America, **5** Department of Medicine, Beth Israel Deaconess Medical Center, Boston, Massachusetts, United States of America, **6** Department of Biostatistics, Harvard T.H. Chan School of Public Health, Boston, Massachusetts, United States of America, **7** Department of Anesthesia, Critical Care and Pain Medicine, Beth Israel Deaconess Medical Center, Harvard Medical School, Boston, Massachusetts, United States of America, **8** Big Data Department, Fundacion Progreso y Salud, Regional Ministry of Health of Andalucia

☯ These authors contributed equally to this work.
‡ LAC and DH also contributed equally to this work.
* maarmeng@mit.edu

**Data Availability Statement:** The data for the high resolution model were extracted from the eICU Collaborative Research Database, a freely available

## Abstract

High resolution clinical databases from electronic health records are increasingly being used in the field of health data science. Compared to traditional administrative databases and disease registries, these newer highly granular clinical datasets offer several advantages, including availability of detailed clinical information for machine learning and the ability to adjust for potential confounders in statistical models. The purpose of this study is to compare the analysis of the same clinical research question using an administrative database and an electronic health record database. The Nationwide Inpatient Sample (NIS) was used for the low-resolution model, and the eICU Collaborative Research Database (eICU) was used for the high-resolution model. A parallel cohort of patients admitted to the intensive care unit (ICU) with sepsis and requiring mechanical ventilation was extracted from each database. The primary outcome was mortality and the exposure of interest was the use of dialysis. In the low resolution model, after controlling for the covariates that are available, dialysis use was associated with an increased mortality (eICU: OR 2.07, 95% CI 1.75–2.44, p<0.01; NIS: OR 1.40, 95% CI 1.36–1.45, p<0.01). In the high-resolution model, after the addition of the clinical covariates, the harmful effect of dialysis on mortality was no longer significant (OR 1.04, 95% 0.85–1.28, p = 0.64). The results of this experiment show that the addition of high resolution clinical variables to statistical models significantly improves the ability to control for important confounders that are not available in administrative datasets. This suggests that the results from prior studies using low resolution data may be inaccurate and may need to be repeated using detailed clinical data.

multi-center database for critical care research. Pollard TJ, Johnson AEW, Raffa JD, Celi LA, Mark RG and Badawi O. Scientific Data (2018). DOI: http://dx.doi.org/10.1038/sdata.2018.178. Available from: https://www.nature.com/articles/sdata2018178 The data for the low resolution model were extracted from the Nationwide Inpatient Sample (NIS), a national, all-payer database: HCUP Databases. Healthcare Cost and Utilization Project (HCUP). Agency for Healthcare Research and Quality, Rockville, MD. Available from: https://www.hcup-us.ahrq.gov/nisoverview.jsp The authors provide open access to all their data extraction, filtering, data wrangling, modeling, figures and tables, code, and queries on https://github.com/theonesp/hr_vs_lr_repos.

**Funding:** Research reported in this publication was supported by the National Institute of Health grants T32DK007527 (ERG) and NIBIB R01EB017205 (LAC). The funders had no role in study design, data collection and analysis, decision to publish, or preparation of the manuscript.

**Competing interests:** I have read the journal's policy and the authors of this manuscript have the following competing interests: Leo Anthony Celi is the Editor-in Chief of PLOS Digital Health.

## Author summary

Healthcare administrative databases and disease registries are frequently used in clinical research; however, these sources of data were often not designed for this purpose and lack important detailed clinical data. Therefore, when using these data to answer clinical research questions, important clinical variables are missing and may bias the results. Over the past decade, high resolution databases that integrate administrative information and clinical patient data obtained from electronic health records have been developed specifically for the purpose of clinical research. The purpose of this study is to compare the effects of dialysis on mortality in similar cohorts of critically ill patients with sepsis requiring mechanical ventilation from both an administrative database and from a high resolution database. We found that the addition of clinical variables significantly altered the mortality odds ratio such that it was no longer significant. These results suggest that previous studies using administrative data and repositories may not be valid due to the lack of important clinical variables included in the models.

## Introduction

Low resolution databases are data sets that lack granular and detailed clinical data, and often contain pre-specified types of information, such as patient demographics, diagnoses, hospital information as well hospital admission and discharge information [1,2]. Administrative databases, which have been utilized for medical research purposes since they were first created in the 1970s, are one example of a low resolution database [3]. These databases have allowed for the analysis of large amounts of healthcare data over the past decades and have been responsible for numerous practice-changing studies [4–6]. Administrative databases, such as the Nationwide Inpatient Sample (NIS), provide large patient samples and include valuable and reliable information such as patient demographics, diagnostic coding of primary and secondary diagnoses, procedures performed, length of hospitalization and discharge status (ie. Discharge, death, transfer to another facility) [7]. These databases are easily accessible, inexpensive and permit the study of practices and outcomes across a large spectrum of healthcare related research questions. However, these databases were often created with the intent of gathering data for financial, health policy or administrative use, and therefore have inherent limitations. Information bias including coding misclassification and coding accuracy may be present and must be carefully evaluated when using these data sources [1,8–10]. Furthermore, most administrative databases lack follow-up information and clinical information, such as patient vital signs, laboratory values and medication use. Therefore, important additional clinical variables and confounders may be unmeasured, leading to bias in the results.

Medical registries are another type of low resolution data, which include health services registries, product registries and disease registries, and are also important sources of data for medical research and epidemiological studies [11–13]. In contrast to medical databases, medical registries are created with specific well-defined characteristics including that entries are unique individual identifiable persons sharing a common feature, the population is geographically defined, the registry has a pre-defined purpose, and the registry is updated systematically [11,14,15]. Medical registries have been used in numerous practice changing studies, and are particularly useful in the study of rare diseases [16–19]. The data collected by registries are variable and determined by the specific objectives of the registry, but are typically sourced from electronic health records (EHRs), medical charts, administrative databases, and patient reports [15,20]. Although medical registries may contain limited specific clinical data, similar to

administrative databases, registries lack granularity. Furthermore, registries are also limited by information biases including coding misclassification, data collection errors, and data completeness errors [15].

More recently, the widespread use of EHRs has provided access to large amounts of clinical data [21,22]. The EHR data with the integration of clinical monitoring systems have allowed for the development of modern databases containing high fidelity clinical information. These datasets with highly granular and detailed clinical data can be considered as high resolution databases. High resolution databases often draw information from EHRs including patient monitoring parameters and physiological data, patient care flow sheets, laboratory values, procedures performed as well as the temporal trends throughout a patient's clinical course [3,23]. Databases such as the eICU Collaborative Research Database (eICU) and the Medical Information Mart for Intensive Care (MIMIC) are examples of high resolution databases [24,25]. These datasets directly integrate clinical data from EHRs and bedside monitoring linearly over time into comprehensive datasets that can be used for clinical research. Utilization of these datasets may allow for better adjustment for confounding as they contain detailed clinical information that administrative datasets and disease registries lack. But the use of datasets such as these and other big data resources are similarly prone to errors and bias, including computational errors with complex statistical techniques, quality and reliability of collected data, and lack of familiarity with knowledge translation to clinical practice [26–28].

Statistical models used to analyze data in non-randomized trials are limited by numerous factors. One of the major barriers to accuracy of modeling is the presence of residual confounding not accounted for by the study design [29]. Specifically, in the intensive care unit, administrative datasets often lack detailed clinical information about severity of illness, laboratory investigations, and vital signs, as well time-stamped interventions, which are likely important confounders not accounted for in the models [30]. Conversely, the use of robust databases that incorporate clinical and bedside monitoring data might improve modelling accuracy due to the ability to control for additional clinical variables and confounders, that are not available with low resolution datasets. The aim of this experiment is to compare the ability to adjust for confounding between a low resolution large national administrative database and a high resolution large multi-center EHR database examining the same clinical question.

## Results

There were 139,367 patients included in the 2014 eICU sample, of which a total of 8,822 (6.3%) patients were included in the cohort (Fig 1). A total of 7,071,762 hospitalizations from the 2014 NIS sample were analyzed and 223,947 (3.2%) were included in the cohort (Fig 2). The overall mortality was 22.6% in the eICU cohort, while the NIS cohort had a mortality of 26.9%. There were 727 (8.2%) patients in the eICU cohort who required dialysis, with a mortality rate of 36.0%; the NIS cohort contained 19,149 (8.5%) patients who required dialysis with a mortality rate of 40.0%. The baseline characteristics of the cohorts are displayed in Table 1. For the high resolution data variables for the eICU cohort, the baseline levels are available in S1 Table.

The results of the low-resolution model for in-hospital mortality are displayed in Table 2. After controlling for all variables in the model, dialysis use was associated with an increased risk of mortality in both cohorts (eICU: OR 2.07, 95% CI 1.75–2.44, p<0.01; NIS: OR 1.40, 95% CI 1.36–1.45, p<0.01).

For the eICU cohort, after addition of the high resolution covariates, the point estimate for the detrimental effect of hemodialysis use was no longer significant (OR 1.04, 95% 0.85–1.28, p = 0.64). The results of the high resolution model are displayed in Table 3. The correlation

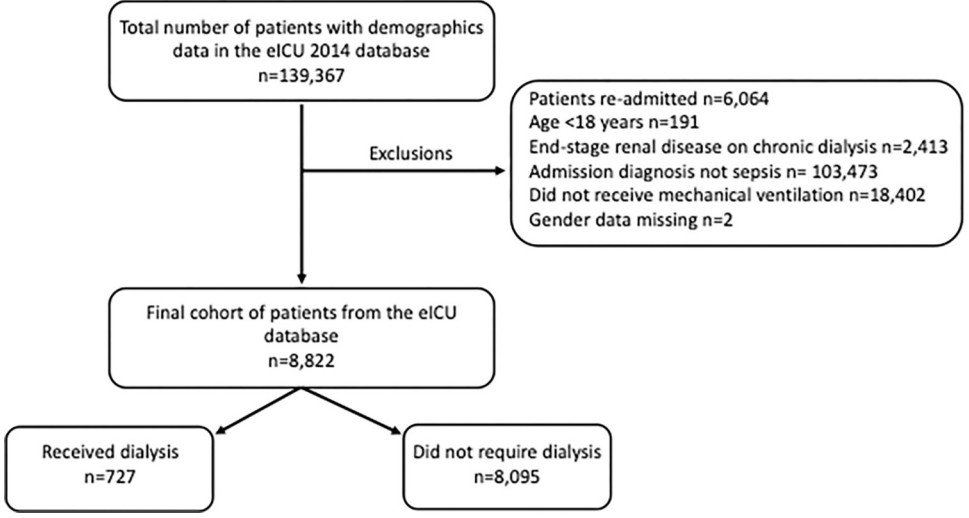

**Fig 1. eICU cohort.** The cohort selection from the eICU database.

plot (Fig 3) shows that overall, there was not significant correlation between the majority of the high resolution variables. The pairs of variables that did show high correlation were hemoglobin and hematocrit, AST and ALT, mean arterial pressure and diastolic blood pressure, and mean arterial pressure and systolic blood pressure.

The authors provide open access to all their data extraction, filtering, data wrangling, modeling, figures and tables, code, and queries on https://github.com/theonesp/hr_vs_lr_repos.

## Discussion

In this comparative analysis utilizing comparable cohorts of mechanically ventilated patients with sepsis, the addition of the high-resolution clinical variables in the eICU database allowed for greater adjustment of severity of illness and significantly altered the point estimate for the

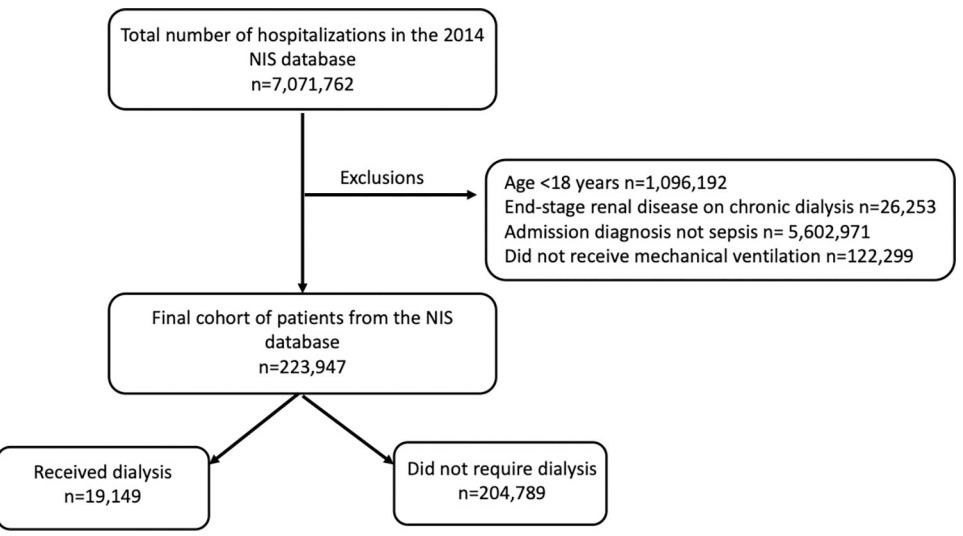

**Fig 2. NIS cohort.** The cohort selection from the NIS database.

**Table 1. The baseline characteristics of the patients in the eICU and NIS cohorts.** The patients receiving dialysis and those not receiving dialysis from each cohort are compared using the Student's independent t-test of Wilcoxon Rank Sum test for continuous variables, and the Chi-Square test for categorical variables.

| | eICU Cohort | | | | NIS Cohort | | | |
|---|---|---|---|---|---|---|---|---|
| | Total Cohort (n = 8,822) | No dialysis (n = 8,095) | Dialysis (n = 727) | p-value | Total Cohort (n = 223,947) | No Dialysis (n = 204,798) | Dialysis (n = 19,149) | p-value |
| **Age, mean (SD)** | 64.2 (16.4) | 64.4 (16.5) | 61.4 (14.9) | <0.01 | 55.0 (24.8) | 54.4 (25.4) | 61.6 (15.2) | <0.01 |
| **Male, n (%)** | 4,688 (53.1) | 4,266 (52.7) | 422 (58.0) | <0.01 | 123,084 (55.0) | 112,135 (54.7) | 10,949 (57.2) | <0.01 |
| **Mortality, n (%)** | 1,996 (22.6) | 1,734 (21.4) | 262 (36.0) | <0.01 | 60,330 (26.9) | 52,667 (25.7) | 7,63 (40.0) | <0.01 |
| **Race, n (%)** | | | | 0.75 | | | | <0.01 |
| White | 6,884 (78.0) | 6324 (78.1) | 560 (77.0) | | 138,714 (61.9) | 128,829 (62.9) | 9,885 (51.6) | |
| Black | 855 (9.7) | 781 (9.6) | 74 (10.2) | | 35,861 (16.0) | 31,208 15.2) | 4,653 (24.3) | |
| Hispanic | 392 (4.4) | 361 (4.5) | 31 (4.3) | | 21,323 (9.5) | 19,076 (9.3) | 2,247 (11.7) | |
| Asian | 126 (1.4) | 117 (1.4) | 9 (1.2) | | 5,570 (2.5) | 4,889 (2.2) | 681 (3.6) | |
| Native American | 62 (0.7) | 54 (0.7) | 8 (1.1) | | 1,588 (0.7) | 1,433 (0.7) | 155 (0.8) | |
| Other/Unknown | 503 (5.7) | 458 (5.7) | 45 (6.2) | | 20,883 (9.3) | 19,355 (9.4) | 1,528 (7.9) | |
| **Region of Country, n (%)** | | | | <0.01 | | | | <0.01 |
| Northeast | 1,072 (12.2) | 993 (12.3) | 79 (10.9) | | 38,824 (17.3) | 35,799 (17.4) | 3,025 (15.8) | |
| Midwest | 2,419 (27.4) | 2,243 (27.7) | 176 (24.2) | | 49,067 (21.9) | 44,672 (21.8) | 4,395 (23.0) | |
| South | 2,753 (31.2) | 2,544 (31.4) | 209 (28.7) | | 91,341 (40.8) | 83,748 (40.9) | 7,593 (39.7) | |
| West | 1,960 (22.2) | 1,742 (21.5) | 218 (30.0) | | 44,715 (20.0) | 40,579 (19.8) | 4,136 (21.6) | |
| Other/Unknown | 618 (7.0) | 573 (7.1) | 45 (6.2) | | | | | |
| **Shock, n (%)** | 2,230 (25.3) | 1,938 (23.9) | 292 (40.2) | <0.01 | 71,016 (31.7) | 60,623 (29.6) | 10,393 (54.2) | <0.01 |
| **Charlson Comorbidity Index, mean (SD)** | 3.90 (2.73) | 3.93 (2.75) | 3.59 (2.51) | <0.01 | 1.2 (1.1) | 1.2 (1.1) | 1.5 (1.0) | <0.01 |
| **Hospital size, n (%)** | | | | <0.01 | | | | <0.01 |
| ≤100 beds | 292 (3.3) | 283 (3.5) | 9 (1.2) | | 31,029 (13.9) | 28,542 (13.9) | 2,487 (13.0) | |
| 100–249 beds | 1,751 (19.8) | 1,632 (20.2) | 119 (16.4) | | 62,304 (27.8) | 57,101 (27.9) | 5,203 (27.2) | |
| ≥250 beds | 5,903 (66.9) | 5,390 (66.6) | 513 (70.6) | | 130,614 (58.3) | 119,155 (91.2) | 11,459 (8.8) | |
| Other/unknown | 876 (9.9) | 790 (9.8) | 86 (11.8) | | - | - | - | - |
| **Teaching hospital, n (%)** | 2,453 (27.8) | 2,226 (27.5) | 227 (31.2) | 0.04 | 161,657 (72.2) | 147,777 (72.1) | 13,880 (72.5) | 0.58 |

association of hemodialysis use and hospital mortality. We demonstrate that the cohorts of patients that were obtained from the low-resolution NIS database and the high resolution eICU-CRD are comparable by baseline patient and hospital demographics, dialysis use, and in-hospital mortality. The results show that the baseline low resolution models from both the cohorts predicted a significant association of in-hospital mortality with dialysis use. This is in keeping with previously published epidemiological studies that have reported that dialysis is associated with increased in-hospital mortality among critically ill patients with sepsis who require mechanical ventilation [31,32]. After adjusting for the high resolution variables in our model, the association of in-hospital mortality and dialysis use in the eICU cohort was no longer significant. These results demonstrate that after the addition of the high resolution clinical variables to the model, the results of the analysis changed significantly. This highlights the importance of the granular clinical data within the high resolution model, as the low resolution model did not account for these important clinical confounders.

Administrative databases and disease registries have been widely used for decades to examine clinical questions in all areas of healthcare, and the publication of such epidemiological studies has increased over time [6]. Often these data sources were not created with the intention of answering clinical questions and lack the detailed clinical information required for proper adjustment in statistical modelling to remove residual confounding [7,10]. The use of

**Table 2. The low resolution multivariable logistic regression model predicting in-hospital mortality in the NIS and eICU cohorts.**

| | eICU cohort | | | NIS cohort | | |
|---|---|---|---|---|---|---|
| | **Odds ratio** | **95% CI** | **p-value** | **Odds ratio** | **95% CI** | **p-value** |
| **Dialysis** | **2.07** | **1.75–2.44** | **<0.01** | **1.4** | **1.36–1.45** | **<0.01** |
| **Age** | 1.01 | 1.01–1.01 | <0.01 | 1.02 | 1.02–102 | <0.01 |
| **Male gender** | 0.93 | 0.84–1.03 | 0.15 | 1.04 | 1.02–1.06 | <0.01 |
| **Race (reference = white)** | | | | | | |
| Black | 1.02 | 0.85–1.22 | 0.81 | 0.99 | 0.96–1.02 | 0.43 |
| Hispanic | 0.82 | 0.63–1.06 | 0.13 | 1.03 | 0.99–1.06 | 0.16 |
| Asian | 0.87 | 0.56–1.36 | 0.54 | 1.17 | 1.10–1.25 | <0.01 |
| Native American | 1.1 | 0.59–2.06 | 0.76 | 1.11 | 0.98–1.25 | 0.09 |
| Other/Unknown | 1.15 | 0.92–1.43 | 0.22 | 1.2 | 1.16–1.24 | <0.01 |
| **Region of Country (reference = South)** | | | | | | |
| Northeast | 1.04 | 0.86–1.26 | 0.7 | 1.01 | 0.99–1.04 | 0.32 |
| Midwest | 0.82 | 0.71–0.95 | 0.01 | 0.86 | 0.84–0.88 | <0.01 |
| West | 1.03 | 0.89–1.19 | 0.75 | 0.97 | 0.94–0.99 | 0.01 |
| Other/Unknown | 0.76 | 0.53–1.09 | <0.01 | - | - | - |
| **Shock** | 1.95 | 1.75–2.18 | <0.01 | 2.65 | 2.59–2.70 | <0.01 |
| **Charlson Comorbidity Index** | 1.11 | 1.08–1.13 | <0.01 | 1.13 | 1.12–1.14 | <0.01 |
| **Hospital Size (reference = <100 beds)** | | | | | | |
| 100–249 beds | 1.22 | 0.89–1.68 | 0.23 | 1.11 | 1.08–1.15 | <0.01 |
| >250 beds | 1.13 | 0.83–1.54 | 0.41 | 1.2 | 1.17–1.24 | <0.01 |
| Other / Unknown | 0.76 | 0.53–1.09 | 0.13 | - | - | - |
| **Teaching Hospital** | 1.15 | 1.00–1.32 | 0.06 | 1.2 | 1.17–1.22 | <0.01 |

administrative databases such as the NIS have inherent limitations related to coding and misclassification bias [33]. Furthermore, a recent study found that up to 85% of studies published using the NIS database did not adhere to the specified methodological standards, which can further bias study results and interpretations [34]. Disease registries, which are non-randomized observational datasets containing patient, medical treatment or device information, are also limited by selection biases, information biases, and data quality errors [35,36]. Furthermore, data registries in particular may be prone to data linkage errors, which can also mislead study findings [37,38]. For example, a recent study evaluating prostate specific antigen values in cancer registries found high rates of misclassification error when compared to the gold standard EHR laboratory value, resulting in important differences in clinical outcomes [39]. Both administrative datasets and disease registries lack detailed clinical information, which are important confounders to consider in clinical research, as study results can be greatly influenced with the addition of these variables.

In contrast to administrative datasets and disease registries, more recently developed high resolution databases such as eICU, were created with its use for clinical research as a primary objective. Globally, there are several other examples of commercial and non-commercial high resolution databases and national repositories in the field of critical care that are currently used for medical research. In the United States, examples include the Veterans Affairs patient database, the MIMIC-II database and the APACHE and Project IMPACT databases [25,40–42]. In the United Kingdom, the Intensive Care National Audit and Research Center (ICNARC) provides high resolution data from national repositories, and similarly, the Australian and New Zealand Intensive Care Society also curates a large database from patient ICU stays [43]. These high resolution datasets are well designed and structured to answer clinical

**Table 3.  The high resolution multivariable logistic regression model predicting in-hospital mortality in the eICU cohort.**

| Variable | Odds ratio | 95% CI | p-value |
|---|---|---|---|
| **Dialysis** | 1.04 | 0.85–1.28 | 0.64 |
| **Age** | 1.02 | 1.01–1.02 | <0.01 |
| **Male gender** | 0.93 | 0.83–1.05 | 0.26 |
| **Race (reference = White)** | | | |
| Black | 0.99 | 0.80–1.23 | 1.00 |
| Hispanic | 0.84 | 0.63–1.12 | 0.24 |
| Asian | 1.09 | 0.67–1.76 | 0.77 |
| Native American | 1.14 | 0.56–2.33 | 0.65 |
| Other/Unknown | 1.18 | 0.92–1.53 | 0.19 |
| **Region of Country (reference = South)** | | | |
| Northeast | 1.22 | 0.98–1.52 | 0.11 |
| Midwest | 0.86 | 0.72–1.02 | 0.06 |
| West | 1.03 | 0.87–1.22 | 0.79 |
| Other/Unknown | 1.70 | 1.31–2.21 | <0.01 |
| **Shock** | 0.87 | 0.75–1.00 | 0.04 |
| **Charlson Comorbidity Index** | 1.07 | 1.04–1.10 | <0.01 |
| **Hospital Size (reference ≤100 beds)** | | | |
| 100–249 beds | 0.91 | 0.64–1.31 | 0.56 |
| ≥250 beds | 0.80 | 0.57–1.14 | 0.43 |
| Other / Unknown | 0.57 | 0.38–0.86 | 0.01 |
| **Teaching hospital status** | 1.15 | 0.98–1.35 | 0.01 |
| **Laboratory tests** | | | |
| Sodium (mEq/L) | 1.00 | 0.99–1.01 | 0.77 |
| Potassium (mEq/L) | 1.07 | 0.98–1.17 | 0.12 |
| Bicarbonate (mEq/L) | 1.00 | 0.98–1.01 | 0.49 |
| Blood urea nitrogen (mg/dL) | 1.00 | 1.00–1.01 | 0.06 |
| Creatinine (mg/dL) | 0.99 | 0.93–1.05 | 0.61 |
| Glucose (mg/dL) | 1.00 | 1.00–1.00 | 0.51 |
| Calcium (mg/dL) | 1.09 | 1.02–1.17 | 0.01 |
| Phosphate (mg/dL) | 1.06 | 1.01–1.11 | 0.01 |
| Hematocrit (%) | 0.97 | 0.94–1.01 | 0.17 |
| Hemoglobin (g/dL) | 1.04 | 0.93–1.16 | 0.48 |
| Red blood cell distribution width (RDW) | 1.05 | 1.02–1.08 | <0.01 |
| Platelet count ($mm^{-3}$) | 1.00 | 1.00–1.00 | <0.01 |
| White blood cell count ($mm^{-3}$) | 1.00 | 1.00–1.01 | 0.24 |
| International Normalized Ratio (INR) | 1.08 | 1.01–1.15 | 0.03 |
| Lactate (mg/dL) | 1.14 | 1.11–1.17 | <0.01 |
| Albumin (g/dL) | 0.81 | 0.73–0.91 | <0.01 |
| Bilirubin total (mg/dL) | 1.04 | 1.02–1.07 | <0.01 |
| Alanine aminotransferase (mU/mL) | 1.00 | 1.00–1.00 | 0.47 |
| Aspartate aminotransferase (mU/mL) | 1.00 | 1.00–1.00 | 0.92 |
| Alkaline phosphatase (U/L) | 1.00 | 1.00–1.00 | 0.13 |
| Lactate dehydrogenase (U/L) | 1.00 | 1.00–1.00 | 0.05 |
| **Vital signs** | | | |
| Fraction of inspired oxygen ($FiO_2$) | 2.55 | 1.70–3.83 | <0.01 |
| Heart rate (bpm) | 1.01 | 1.00–1.01 | <0.01 |

(*Continued*)

**Table 3.** (Continued)

| Variable | Odds ratio | 95% CI | p-value |
|---|---|---|---|
| Respiratory rate (min$^{-1}$) | 1.03 | 1.02–1.04 | <0.01 |
| Oxygen saturation (SpO$_2$, %) | 0.91 | 0.89–0.93 | <0.01 |
| Systolic blood pressure (mmHg) | 0.99 | 0.99–1.00 | 0.01 |
| Diastolic blood pressure (mmHg) | 1.00 | 0.99–1.01 | 0.46 |
| Mean Arterial Pressure (mmHg) | 1.00 | 0.98–1.01 | 0.36 |
| Temperature (C) | 0.76 | 0.71–0.82 | <0.01 |
| **Medication use (reference = not used)** | | | |
| Any vasopressor | 2.30 | 2.00–2.65 | <0.01 |
| Dopamine | 1.02 | 0.76–1.38 | 0.90 |
| Dobutamine | 1.02 | 0.70–1.49 | 0.89 |
| Norepinephrine | 0.98 | 0.84–1.14 | 0.76 |
| Phenylephrine | 1.37 | 1.12–1.67 | <0.01 |
| Epinephrine | 1.59 | 1.19–2.14 | <0.01 |
| Vasopressin | 1.13 | 0.93–1.37 | 0.20 |
| Milrinone | 0.92 | 0.39–2.14 | 0.89 |
| Heparin | 1.33 | 1.03–1.72 | 0.02 |

questions. They also contain vast amounts of clinical data such as patient vital signs and laboratory values captured from various electronic sources in hospitals [24]. These datasets are increasingly being used due to the powerful data available to address clinical research questions, and have provided data to support hundreds of studies in recent years. However, given the large amount of data drawn from multiple sources within these databases, data integrity and accuracy of the collected data are very important considerations when performing analyses because systematic errors in the data lead to propagated error within models and may influence the conclusions of the analysis [44,45]. While data integrity is also a concern in administrative datasets, the sheer volume of data collated in high resolution databases may compound the degree of error. As our experiment demonstrates, the inclusion of detailed clinical variables in the model leads to a difference in research conclusions compared to when using the administrative database. This is despite baseline similarities between the two cohorts with respect to patient demographics, dialysis use and clinical outcomes. Not surprisingly, this supports the argument that important confounders are lacking in administrative databases, and therefore conclusions drawn from prior research using these types of databases may not be accurate.

The strengths of this analysis lie in the large sample sizes and multi-center scope of both datasets [24,46]. Our analysis is generalizable to a significant proportion of the critical care population from across the United States, taking into account the limitations of each dataset. The accuracy of coding for dialysis use has also been shown to be highly reliable by prior studies, and the diagnostic coding has been validated for use in administrative databases [47].

The results of the study must be interpreted while acknowledging several important limitations. As described above, the use of administrative and EHR based datasets contains inherent risks for data error, misclassification bias and coding errors. The NIS database does not identify individual patients, but rather each entry is a patient encounter [48]. Accordingly, it is possible that if a patient is transferred between hospitals, this may represent more than one entry in the database. Limitations with the eICU database are inherent in the fact that the data are derived from multiple eICUs across the United States. A recent study by O'Halloran and colleagues showed that the eICU database may lack generalizability to large ICUs with more acuity, as the majority of data from within the database is obtained from small and medium sized

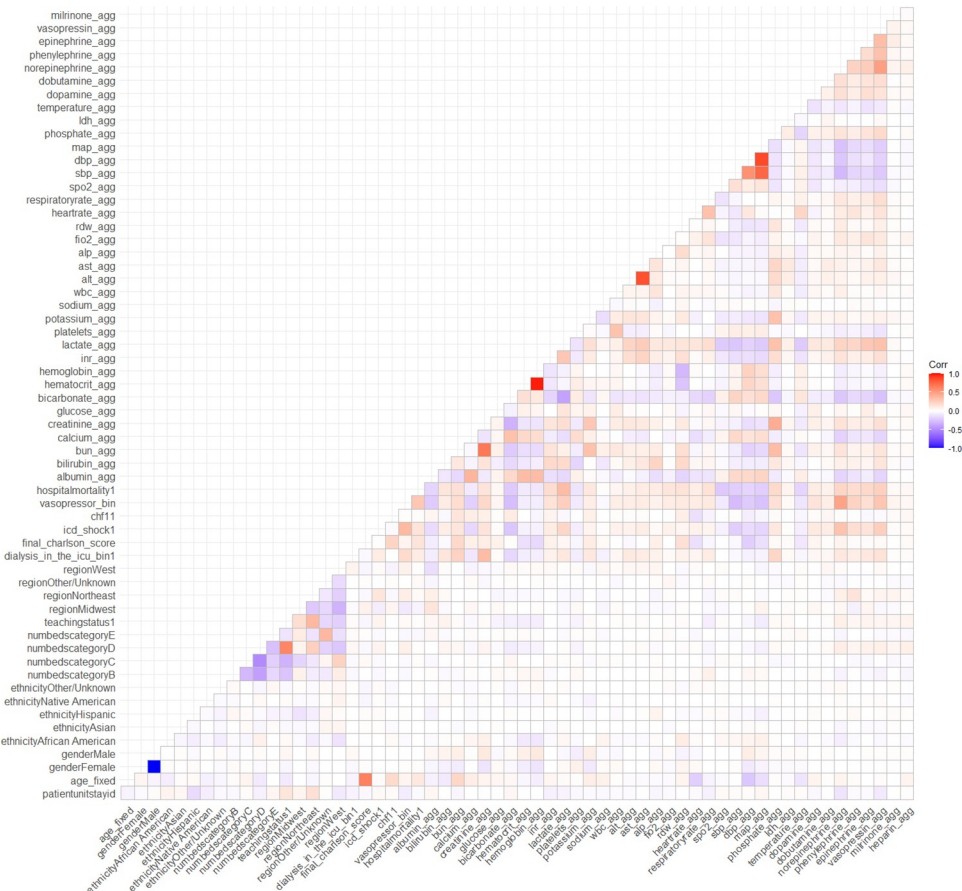

**Fig 3. The correlation plot for the variables in the high resolution model.**

ICUs [49]. Furthermore, this study identified potential ambiguity with coding of mechanical ventilation within the dataset therefore this may have influenced our cohort selection [49].

This study serves as a thought experiment and proof of concept that highlights the benefits high-resolution data sources from EHRs have over low-resolution administrative datasets. Our findings show that the addition of granular clinical data to administrative data elements significantly altered the point estimate for the association of dialysis use and mortality in patients with sepsis. The results of this experiment suggest that many prior analyses utilizing low resolution administrative data may need to be repeated with more powerful data sources in order to better assess causal relationships while controlling for important clinical confounders. Another avenue that is worth exploring is the linking of multi-center EHR datasets with claims and registry data to combine the benefits of these different types of data sources. Further studies examining the outcome of clinical research questions using both high and low resolution datasets should be performed in the future to further evaluate the impact of dataset granularity on the results of observational studies.

## Materials and methods

### Study rationale

For this experiment we attempted to create comparable cohorts of patients in each dataset with inclusion criteria that would allow for a high degree of certainty. Mechanical ventilation

and the use of hemodialysis are well coded in administrative datasets, whereas sepsis is a common indication for admission to the intensive care unit.

The Nationwide Inpatient Sample (NIS) was selected to represent the low resolution administrative dataset [48]. This national, all-payer database is produced by the United States Agency for Healthcare Quality and Research (AHRQ). It captures approximately 20% of all inpatient hospitalizations and is designed to approximate >95% of all inpatient care (prison hospitals and non-traditional hospitals are excluded). It is a well-validated database that has excellent data integrity and has been used for decades for health services research. The eICU Collaborative Research Database (eICU-CRD) was selected to represent the high resolution dataset [24]. This eICU-CRD is a publicly available database curated from a partnership between the Laboratory for Computational Physiology (LCP) at Massachusetts Institute of Technology (MIT) and the Electronic ICU Research Institute. The eICU database contains the high granularity de-identified data for over 200,000 inpatient admissions to telehealth ICUs across the United States. This database has been used for hundreds of research projects since it became publicly available in 2017, and the data has undergone stringent technical validation [24]. The use of the eICU-CRD is exempt from institutional review board approval due to the retrospective design, lack of direct patient intervention, and the security schema, for which the re-identification risk was certified as meeting safe harbor standards by an independent privacy expert (Privacert, Cambridge, MA) (Health Insurance Portability and Accountability Act Certification no. 1031219–2).

A waiver of consent for this analysis was obtained from the Research Ethics Board at the University of Manitoba, as all of the data is publicly available and de-identified. This study is reported in accordance with the STrengthening the Reporting of OBservational studies in Epidemiology (STROBE) statement [50].

## Cohort selection

Comparable cohorts of patients with sepsis who required mechanical ventilation were created in the NIS and eICU databases. The details of cohort selection for each database can be found in Figs 1 and 2. The baseline characteristics of the patients in both the NIS and eICU cohorts is shown in Table 1. The patients from each cohort are compared by use of dialysis using either the Student's Independent t-test or the Wilcoxon Rank Sum depending on normality for continuous variables, and the Chi-Square analysis for categorical variables. All statistical analyses in this study were performed assuming a two-sided alpha level of 0.05.

## Base Covariates (eICU and NIS)

For each dataset, a baseline set of common variables was collected. Covariates included age, gender, race (White, Black, Hispanic, Other/Missing), Region of Country (West, Northeast, South, Midwest, Missing), Hospital size (<100 beds, 100–249 beds, $\geq$ 250 beds, other/unknown), Charlson comorbidity index, the presence of shock, teaching status of hospital, and the use of dialysis.

## High Resolution Covariates (eICU only)

Additional high-resolution covariates were obtained for each patient from the eICU-CRD. These included the patient laboratory results (sodium, potassium, bicarbonate, blood urea nitrogen (BUN), creatinine, glucose, calcium, phosphate, hematocrit, hemoglobin, red cell distribution width (RDW), platelet count, white blood cell count, international normalized ratio (INR), lactate, liver enzymes and function tests), patient vital signs (fraction of inspired oxygen ($FiO_2$), heart rate, respiratory rate, oxygen saturation, blood pressure and temperature) as well

as medication use (any vasopressor or inotrope including dopamine, dobutamine, norepinephrine, phenylephrine, epinephrine, vasopressin, milrinone and heparin). For variables with more than one record per hour, a median value per hour was computed. The hourly average was then determined and aggregated for each variable per patient. Missing data for the high-resolution model continuous variables were imputed using a forward-backward filling imputation method. Forward filling means to fill missing values with previous data available for a given patient. Backward filling means to fill missing values with the next data point available for a given patient. The function first attempts to fills the data point with the backward method if a datapoint is available, and if not available, then with the forward method. For binary data, missing data was treated as a 0. Missing data for the high-resolution binary medication variables were treated as not administered. The plot of missingness for each variable is shown in supplemental material S1 Fig.

### Low resolution model

In order to compare the output of the two datasets, identical multivariate logistic regression models predicting mortality after ICU admission for sepsis while requiring mechanical ventilation were created. The variables included in the baseline low-resolution model, which were the same for both datasets, were chosen *a priori* and are the same variables described in the *Baseline Covariates* section of the Methods. Normalization of the variables was not performed prior to performing the logistic regression analysis.

### High resolution model

A multivariate logistic regression model (estimated using maximum likelihood) was fitted to predict in-hospital mortality for patients admitted to the ICU with sepsis while requiring mechanical ventilation was performed with the eICU cohort. Standardized parameters were obtained by fitting the model on a standardized version of the dataset. A typical predictor is fitted for binomial families, the response is specified as a factor where the first level denotes failure and all others success. Both the low resolution (listed in the Base Covariates section) and high-resolution (listed in the High Resolution Covariates section) covariates described above were used, and were not normalized prior to analysis. The variables included in the high resolution model were determined *a priori*. In order to address collinearity of the model, a correlation plot was generated to study all the interaction between the model variables (Fig 3).

## Supporting information

**S1 Table. The eICU high resolution variables. The baseline characteristics of the high resolution variables for the patients in the eICU database.**
(PDF)

**S1 Fig. The plot of missingness for the high resolution variables.**
(TIF)

## Author Contributions

**Conceptualization:** Barret N. M. Rush, Eric R. Gottlieb, Leo Anthony Celi.

**Data curation:** Miguel Ángel Armengol de la Hoz.

**Formal analysis:** Barret N. M. Rush, Miguel Ángel Armengol de la Hoz.

**Investigation:** Barret N. M. Rush, Eric R. Gottlieb, Leo Anthony Celi, Miguel Ángel Armengol de la Hoz.

**Methodology:** Jennifer Ziegler, Barret N. M. Rush, Eric R. Gottlieb, Leo Anthony Celi, Miguel Ángel Armengol de la Hoz.

**Project administration:** Barret N. M. Rush, Leo Anthony Celi, Miguel Ángel Armengol de la Hoz.

**Resources:** Jennifer Ziegler, Barret N. M. Rush, Leo Anthony Celi.

**Software:** Barret N. M. Rush.

**Supervision:** Barret N. M. Rush, Leo Anthony Celi.

**Validation:** Barret N. M. Rush, Eric R. Gottlieb, Leo Anthony Celi.

**Visualization:** Miguel Ángel Armengol de la Hoz.

**Writing – original draft:** Jennifer Ziegler, Barret N. M. Rush, Miguel Ángel Armengol de la Hoz.

**Writing – review & editing:** Jennifer Ziegler, Barret N. M. Rush, Leo Anthony Celi.

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
