## [Decision Letter · Decision Letter 0]

7 Mar 2022

PDIG-D-21-00095

High resolution data modifies intensive care unit dialysis outcome predictions as compared with low resolution administrative data set

PLOS Digital Health

Dear Dr. Armengol de la Hoz,

Thank you for submitting your manuscript to PLOS Digital Health. After careful consideration, we feel that it has merit but does not fully meet PLOS Digital Health's publication criteria as it currently stands. Therefore, we invite you to submit a revised version of the manuscript that addresses the points raised during the review process.

We look forward to receiving your revised manuscript.

Kind regards,

Gilles Guillot

Academic Editor

PLOS Digital Health

Journal Requirements:

1. Please amend your detailed Financial Disclosure statement. This is published with the article, therefore should be completed in full sentences and contain the exact wording you wish to be published.

i). State the initials, alongside each funding source, of each author to receive each grant.

ii). State what role the funders took in the study. If the funders had no role in your study, please state: “The funders had no role in study design, data collection and analysis, decision to publish, or preparation of the manuscript.”

2. Please update your Competing Interests statement. If you have no competing interests to declare, please state: “The authors have declared that no competing interests exist.”

3. We have noticed that you have uploaded supporting information but you have not included a list of legends. Please add a full list of legends for all supporting information files (including figures, table and data files) after the references list.

Additional Editor Comments (if provided):

The manuscript has been assessed by three reviewers. Their conclusions are moderately enthusiastic. To the relatively long list of comments, I would add that from a strictly statistical perspective, the result boils down to a trivial fact: the estimate and significance of a variable coefficient depends on the model in which the variable is included. I would like to see a more thorough biomedical interpretation of what is taking place here. Is it reverse causation (patients with poorer conditions being prescribed dialysis) or an effect of model over fitting with the high resolution data?

Another important point made by the reviews is that prediction models in ICU's are nearly always based on detailed EHR data not administrative data. It is therefore unclear how valuable the performance difference in models would be for actual clinical use.

The authors are welcome to submit a revision but should be aware that the revised version would have to receive much stronger support from the reviewers to reach threshold for publication. Based on the current version, this will require major improvements.

Reviewers' comments:

Reviewer's Responses to Questions

**Comments to the Author**

1. Does this manuscript meet PLOS Digital Health’s publication criteria? Is the manuscript technically sound, and do the data support the conclusions? The manuscript must describe methodologically and ethically rigorous research with conclusions that are appropriately drawn based on the data presented.

Reviewer #1: Yes

Reviewer #2: No

Reviewer #3: Yes

2. Has the statistical analysis been performed appropriately and rigorously?

Reviewer #1: Yes

Reviewer #2: No

Reviewer #3: Yes

3. Have the authors made all data underlying the findings in their manuscript fully available (please refer to the Data Availability Statement at the start of the manuscript PDF file)?

Reviewer #1: Yes

Reviewer #2: No

Reviewer #3: Yes

4. Is the manuscript presented in an intelligible fashion and written in standard English?

Reviewer #1: Yes

Reviewer #2: Yes

Reviewer #3: Yes

5. Review Comments to the Author

Reviewer #1: This paper compared low resolution clinical dataset (such as NIS) with high resolution clinical dataset (such as eICU) by creating comparable cohorts of patients (who had sepsis and required ventilation). Baseline variables that are available in both datasets were used to build low-resolution models. Baseline variables with extra detailed variables that are unique in the eICU were used to build a high-resolution model. Odds ratio derived from logistic regression model in predicting mortality was used to compare the change of variables' effect in the outcome prediction. 

This study highlights the importance of creating and maintaining high-resolution, high-quality dataset for clinical research. It provides detailed comparison between low-resolution and high-resolution datasets. 

1. For the high-resolution covariates, what was the time range used to predict mortality? e.g. was the median of first hour values after ICU admission used? or was the median of first day values used? 

2. Please specify if normalization of variables was used before doing logistic regression.

3. The p-values in Table 1 are comparison between no-dialysis and with-dialysis in each cohort. Please indicate this in the caption. Also, if the caption could detail the test methods used, that could be better. 

4. Is it necessary to evaluate how good the model fitting was? The experiment shows that no evidence supports that 'dialysis' in the high-res model is significant, while it was in the low-res models. But why would we trust the high-res model? Was it over-fitted? It had much more variables. 

5. It seems that the experiment did not intent to make the populations of the two cohorts similar. Table 1 did not compare between the two cohorts. e.g. the male percentage, the mortality rate and shock in two cohorts seem to be different. There was no difference in the percentage of dialysis (with proportion test). NIS cohort and eICU cohort had similar percentage of cases with dialysis; NIS cohort had higher mortality rate, and shock. Would the population difference affect the model comparison?

Generally, I agree with the authors' claim that more detailed clinical variables would better help to answer clinical questions. With more useful variables being added to the model, it is likely that previously significant variables would be less important. As the authors said, this is "a thought experiment and proof of concept" study. More experiments and comparison in the future would better support the claim.

Reviewer #2: Predictive models in critical care have been around for decades. Almost all of the major ones use information extracted electronically or manually from Electronic Medical Record systems (EMR). It is rare to see an observational study in critical care that uses administrative data, with the only major caveat being that because of the COVID pandemic there were studies that relied on data not gleaned from an EMR. 

It has been well documented that using an administrative database is less precise than using EMR data. Perhaps the novelty in the manuscript by Ziegler et al lies in showing that it’s not just general mortality predictions that are less precise when developed from administrative data, but that clinical studies could have considerably different results based on the data source (administrative vs. EMR). But to effectively make that case necessitates looking at more than one effect-outcome analysis. Thus, my major complaint about this manuscript is that it needs to incorporate more analyses to make its case. Certainly with the NIS and eICU datasets this is not too difficult to carry out.

The authors seem to disregard databases such as APACHE and Project IMPACT, as well as national repositories (ICNARC, ANZICS) that have provided hundreds of clinical studies. These data sources need to be discussed in the manuscript, as they are similar to the eICU data repository in many respects.

There is virtually no information given on the multivariable logistic regression model. In fact, there’s no Methods section at all. Advising readers to go to github rather than present the gist of their methodology in the paper is regretful.

Reviewer #3: I enjoyed reading this article, and I appreciate its perspective. But I think that there are a few considerations that require clarification or editing.

Generally, the article is well written, cogent, and reasonable. However, I found the order to be confusing. The article jumped from introduction to results and then to discussion. It presented Materials and Methods at the end. I was left to read a discussion about a set of models I hadn't yet been introduced to. I think the paper would be greatly improved if it followed a more traditional Introduction, Methods, Results, and Discussion structure. It is also worth noting that there were a large number of comma errors in the paper which distracted me from the content of the paper. These should be corrected.

In the introduction, I struggled with a few points. For example, I didn't understand what the authors meant by "unselected patient samples" (line 97). Additionally, the authors refer to "outcomes" (link 99) but I think this deserves greater clarification. What is the difference between "accuracy and validity" (line 104)? It also isn't clear where "medical registries" fit into this dynamic between low resolution clinical databases vs EHRs. Moreover, it isn't entirely clear to me what the authors mean when they distinguish between low resolution and high resolution in the paper. This must be made more explicit, with consistent usage throughout the paper.

Finally, I found the use of the term "causal inference techniques" (line 136) to be insufficiently vague.

In the Results, it wasn't clear to me how the p-values were calculated. I didn't see sufficient documentation in the methods section to find an answer. This will need to be clarified. Additionally, the code repository should be linked directly (line 181). I was unable to find the exact code in the link provided.

In the Discussion section, I think the core argument of the paper is hidden in line 231. It is my opinion that the paper would benefit from making this more explicit earlier in the paper -- perhaps in the introduction. As further evidence that this is secretly the core argument of the paper, I point to line 254, where the authors clearly document the value of the paper as a contrast between data from EHRs vs administrative databases. But that perspective isn't forcefully presented to the reader, which weakens its value.

In the Methods section, I don't know if "specificity" is the best word choice in line 267. In line 273, I don't think the authors are being entirely clear when they write "and is designed to capture approximately >95% of all inpatient care." How can the database simultaneously "capture" 20% and 95% of inpatient cases? On line 316, the authors write that missing data were imputed using "forward-backward filling imputation method." This will need to clarified. More specifically, under which circumstances did they use forward filling and which circumstances did they use backward filling? For what variables? What percentage of observations were affected? Why do they think this was appropriate over other methods? 

Finally, I think that the authors need to convince the reader that they simply haven't introduced colliders or redundant (i.e. collinear) variables into the models to achieve the demonstration of confounding. If they simply added collinear variables, then, of course, we would see a drop in the odds ratio. How do we know that this isn't the effect in question?

6. PLOS authors have the option to publish the peer review history of their article (what does this mean?). If published, this will include your full peer review and any attached files.

**Do you want your identity to be public for this peer review?** For information about this choice, including consent withdrawal, please see our Privacy Policy.

Reviewer #1: No

Reviewer #2: No

Reviewer #3: No

---

## [Decision Letter · Decision Letter 1]

11 Jul 2022

PDIG-D-21-00095R1

High resolution data modifies intensive care unit dialysis outcome predictions as compared with low resolution administrative data set

PLOS Digital Health

Dear Dr. Armengol de la Hoz,

Thank you for submitting your manuscript to PLOS Digital Health. After careful consideration, we feel that it has merit but does not fully meet PLOS Digital Health's publication criteria as it currently stands. Therefore, we invite you to submit a revised version of the manuscript that addresses the points raised during the review process.

Please submit your revised manuscript within 30 days Sep 09 2022 11:59PM. If you will need more time than this to complete your revisions, please reply to this message or contact the journal office at digitalhealth@plos.org. Please include the following items when submitting your revised manuscript:

We look forward to receiving your revised manuscript.

Kind regards,

Gilles Guillot

Academic Editor

PLOS Digital Health

Journal Requirements:

Additional Editor Comments (if provided):

Your revision addresses the comments brought up during the first review round and we will be happy to accept the manuscript after minor editings suggested by the last review.

Reviewers' comments:

Reviewer's Responses to Questions

**Comments to the Author**

1. If the authors have adequately addressed your comments raised in a previous round of review and you feel that this manuscript is now acceptable for publication, you may indicate that here to bypass the “Comments to the Author” section, enter your conflict of interest statement in the “Confidential to Editor” section, and submit your "Accept" recommendation.

Reviewer #1: All comments have been addressed

2. Does this manuscript meet PLOS Digital Health’s publication criteria? Is the manuscript technically sound, and do the data support the conclusions? The manuscript must describe methodologically and ethically rigorous research with conclusions that are appropriately drawn based on the data presented.

Reviewer #1: Yes

3. Has the statistical analysis been performed appropriately and rigorously?

Reviewer #1: Yes

4. Have the authors made all data underlying the findings in their manuscript fully available (please refer to the Data Availability Statement at the start of the manuscript PDF file)?

Reviewer #1: Yes

5. Is the manuscript presented in an intelligible fashion and written in standard English?

Reviewer #1: Yes

6. Review Comments to the Author

****

Reviewer #1: One more suggestion. Please add one or two sentences to describe the points of Figure 3 (i.e. what do the authors want to tell the readers?). Does it mean only a few variables are high correlated? or all pairwise correlations are under a certain threshold? The plot is just a tool to check, but not a (direct) tool that reduce collinearity.

7. PLOS authors have the option to publish the peer review history of their article (what does this mean?). If published, this will include your full peer review and any attached files.

**Do you want your identity to be public for this peer review?** For information about this choice, including consent withdrawal, please see our Privacy Policy.

Reviewer #1: No

---

## [Editor Report · Decision Letter 2]

9 Sep 2022

High resolution data modifies intensive care unit dialysis outcome predictions as compared with low resolution administrative data set

PDIG-D-21-00095R2

Dear Dr. Armengol de la Hoz,

We are pleased to inform you that your manuscript 'High resolution data modifies intensive care unit dialysis outcome predictions as compared with low resolution administrative data set' has been provisionally accepted for publication in PLOS Digital Health.

Best regards,

Hamish S Fraser, MBCHB MSc

Section Editor

PLOS Digital Health